# Feasibility of comparing medical management and surgery (with neurosurgery or stereotactic radiosurgery) with medical management alone in people with symptomatic brain cavernoma – protocol for the Cavernomas: A Randomised Effectiveness (CARE) pilot trial

James J M Loan [1,2] Andrew Bacon,[3] Janneke van Beijnum,[4]
Pragnesh Bhatt [5] Anna Bjornson [6] Nicole Broomes,[7] Alistair Bullen [8]
Diederik Bulters [7] Julian Cahill [9] Emmanuel Chavredakis [10]
Francesca Colombo [11] Mihai Danciut,[6] Ronneil Digpal,[7]
Richard J Edwards [12] Lucie Ferguson [13] Laura Forsyth,[8] Ioannis Fouyas,[2]
Vijeya Ganesan [14] Patrick Grover [15] Nihal Gurusinghe [11]
Peter S Hall [16] Kirsty Harkness,[3] Lauren S Harris,[17] Tom Hayton,[18]
Adel Helmy [19,20] Daniel Holsgrove,[21] Peter J Hutchinson [19,20] Anil Israni,[22]
Elaine Kinsella,[8] Steff Lewis [8] Sohail Majeed,[5] Conor Mallucci [22]
Nitin Mukerji,[13] Ramesh Nair,[23] Aileen R Neilson [8] Marios C Papadopoulos [24]
Matthias Radatz,[9] Alex Rossdeutsch,[3] Saba Raza-Knight,[11] Jacqueline Stephen,[8]
Andrew Stoddart [8] Mario Teo [25] Carole Turner [19,20] Julia Wade [26]
Daniel Walsh [27,28] David White,[29] Phil White,[30] Jack Wildman,[25]
Oliver Wroe Wright,[27] Christopher Uff [31] Shungu Ushewokunze,[32]
Raghu Vindlacheruvu,[17] Neil Kitchen,[15] Rustam Al-Shahi Salman [1,2,8] on behalf
of the Cavernomas A Randomised Effectiveness (CARE) pilot trial collaborators

For numbered affiliations see end of article.

**Correspondence to**
Professor Rustam Al-Shahi Salman;
rustam.al-shahi@ed.ac.uk

## ABSTRACT

**Introduction** The top research priority for cavernoma, identified by a James Lind Alliance Priority setting partnership was 'Does treatment (with neurosurgery or stereotactic radiosurgery) or no treatment improve outcome for people diagnosed with a cavernoma?' This pilot randomised controlled trial (RCT) aims to determine the feasibility of answering this question in a main phase RCT.

**Methods and analysis** We will perform a pilot phase, parallel group, pragmatic RCT involving approximately 60 children or adults with mental capacity, resident in the UK or Ireland, with an unresected symptomatic brain cavernoma. Participants will be randomised by web-based randomisation 1:1 to treatment with medical management and with surgery (neurosurgery or stereotactic radiosurgery) versus medical management alone, stratified by prerandomisation preference for type of surgery. In addition to 13 feasibility outcomes, the primary clinical outcome is symptomatic intracranial haemorrhage or new

## STRENGTHS AND LIMITATIONS OF THIS STUDY

⇒ Extensive patient, carer and public involvement in the prioritisation of the study question, protocol design, study oversight, support for participants and understanding of barriers to participation.

⇒ A QuinteT recruitment intervention will identify facilitators and barriers to recruitment to inform study materials and recommendations for the method of approach by investigators.

⇒ Participants and investigators will not be blinded to treatment allocation, so there is a risk of non-adherence and performance bias, but blinded outcome adjudication will minimise detection bias.

persistent/progressive focal neurological deficit measured at 6 monthly intervals. An integrated QuinteT Recruitment Intervention (QRI) evaluates screening logs, audio recordings of recruitment discussions, and interviews with

recruiters and patients/parents/carers to identify and address barriers to participation. A Patient Advisory Group has codesigned the study and will oversee its progress.

**Ethics and dissemination** This study was approved by the Yorkshire and The Humber—Leeds East Research Ethics Committee (21/YH/0046). We will submit manuscripts to peer-reviewed journals, describing the findings of the QRI and the Cavernomas: A Randomised Evaluation (CARE) pilot trial. We will present at national specialty meetings. We will disseminate a plain English summary of the findings of the CARE pilot trial to participants and public audiences with input from, and acknowledgement of, the Patient Advisory Group.

**Trial registration number** ISRCTN41647111.

## INTRODUCTION

Symptomatic brain cavernomas are diagnosed in approximately 160 people in the UK annually and cause intracranial haemorrhage and epilepsy.[1–3] Systematic reviews of surgical treatments for cavernomas identified only observational studies.[4–8] These demonstrate that both medical and surgical treatments have risks and benefits.[4–8] No observational study at low risk of bias demonstrates a strong association between surgical treatment and outcome. A randomised controlled trial (RCT) is therefore required to determine whether treatment with neurosurgery or stereotactic radiosurgery (SRS) improves outcome, compared with medical management alone, for patients with symptomatic brain cavernoma.[9] We aim to conduct Cavernomas: A Randomised Effectiveness (CARE) pilot trial to address this. This paper is a published summary of the full protocol (online supplemental material 1).

### Objectives

The primary objective is to assess the feasibility of performing a definitive main phase of an RCT comparing medical management and surgery (with neurosurgery or SRS) versus medical management alone for improving outcome for people with symptomatic brain cavernoma. Secondary objectives are: (1) to set up a collaborative network of patient advocacy organisations and professional representatives at neuroscience centres in the UK and Ireland; (2) to understand recruitment processes and barriers and optimise informed consent and recruitment as part of a QuinteT Recruitment Intervention (QRI) and (3) conduct the CARE pilot trial for approximately 60 people with symptomatic brain cavernoma.

## METHODS AND ANALYSIS

### Design

Two-arm, parallel group randomised pilot trial and feasibility study with an integrated QRI comparing medical management and surgery versus medical management alone, stratified by preferred type of surgical management (figure 1).

### Setting

Participants will be recruited in secondary care settings in the UK and Ireland, from a collaborative network

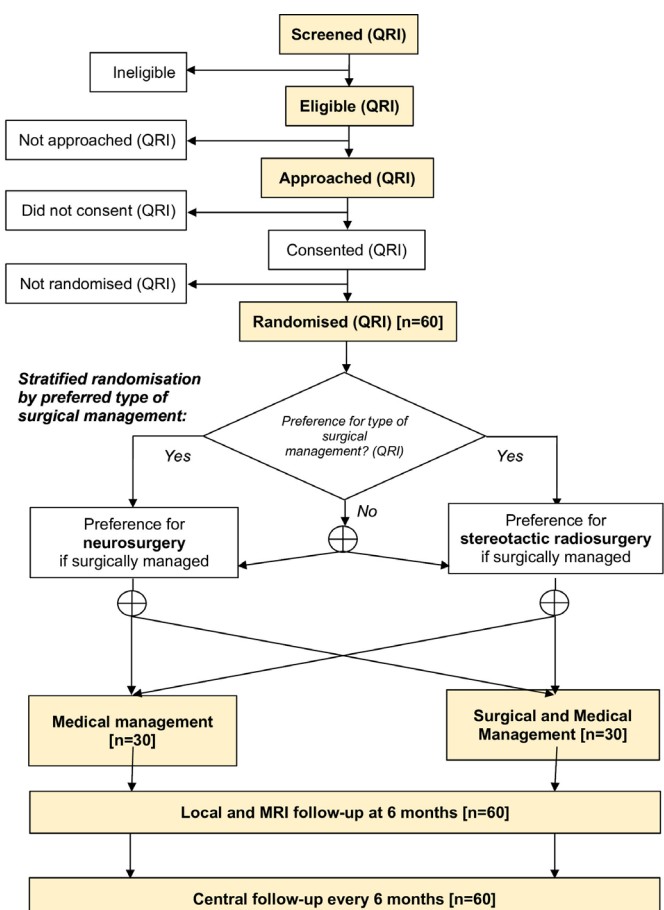

**Figure 1** Participant flow diagram. ⊕, randomised 1:1 allocation; QRI, evaluated by QuinteT Recruitment Intervention.

of research sites. Neurosurgery and follow-up will be conducted by regional neuroscience centres in the UK and Ireland. SRS will be performed at the National Centre for Stereotactic Radiosurgery in Sheffield or the Queen Square Radiosurgery Centre.

### Patient and public involvement

The research question was developed by a priority setting partnership with the patient advocacy organisation Cavernoma Alliance UK (CAUK).[10] A Patient, carer and public Advisory Group (PAG) guided and approved study design and conduct. CAUK will share study information and direct patients to CARE pilot trial sites or to their clinician. Patients will be invited to interviews to explore participation and non-participation decisions. We will disseminate a plain English summary of the study findings to participants and public audiences. We will offer to present our project to annual CAUK meetings.

### Eligibility
#### Inclusion criteria

1. People of any age.
2. At least one brain cavernoma diagnosed by brain MRI that included a gradient echo or susceptibility-

weighted sequence, according to standard diagnostic criteria.[11 12]

3. Clinical history attributable to a brain cavernoma of:[13 14]
   a. Symptomatic stroke due to haemorrhage.
   b. Symptomatic stroke due to a persistent or progressive non-haemorrhagic, or not otherwise specified, focal neurological deficit.
   c. Epileptic seizure(s) meeting the definition of definite or probable cavernoma-related epilepsy.
4. Patient and doctor are uncertain about medical management or medical and surgical management of the symptomatic brain cavernoma, following consultation with a neurosurgeon.
5. Patient has mental capacity to consent for themselves (adult participants or paediatric participants with capacity) or parent/legal guardian provides consent (paediatric participants).

There is no time limit on when a patient may be recruited following the symptomatic presentation and diagnosis of a brain cavernoma. Patients who have previously received surgical management may be included so long as the symptomatic brain cavernoma has not been completely removed/obliterated.

### Exclusion criteria
1. Surgical management of a solitary symptomatic brain cavernoma with MRI evidence of cavernoma removal/obliteration.
2. Spinal cavernoma alone, without symptomatic brain cavernoma.
3. Asymptomatic brain cavernoma. Patients with radiographic cavernoma enlargement (with or without intralesional haemorrhage) but without new symptoms attributable to the cavernoma are still regarded as asymptomatic.
4. Previously randomised in the CARE pilot trial.

### Co-enrolment
Inclusion in another RCT or observational study does not preclude participation in the CARE pilot trial as long as: participants are not overburdened; their inclusion would be unlikely to confound the CARE pilot trial's results or complicate attribution of serious adverse events (SAEs) and outcomes; the protocol of the other study does not preclude co-enrolment in the CARE pilot trial; and co-enrolment has been agreed with the chief investigators (CIs) of all studies involved in co-enrolment.

### Interventions
Patients randomised to medical and surgical management will receive neurosurgical excision or Gamma Knife SRS for their brain cavernoma, in addition to medical management (see comparator), according to what is available in standard clinical practice in the participant's health service.

### Neurosurgical excision
Surgery will be undertaken by a consultant neurosurgeon who will be responsible for neurosurgical aspects of clinical care of that patient in CARE. The neurosurgical technique to resect the cavernoma, including any operative adjuncts, will be that used by that consultant neurosurgeon in usual clinical practice and tailored to each patient according to the consultant neurosurgeon's discretion. Postoperative MRI scan performed within 72 hours of surgery is recommended, but not mandated, to confirm resection completeness.

### Stereotactic radiosurgery
Standard clinical treatment protocols will be used to target the brain cavernoma but not surrounding haemosiderin. Treatment dosages will range from 12 to 16 Gy depending on the size, shape, definition and site of the cavernoma. If intracerebral haemorrhage has occurred from the cavernoma, radiosurgery will be performed once the haematoma is judged to have been reabsorbed to minimise radiation exposure and treatment volume.

### Comparator
Medical management constitutes standard medical care for brain cavernoma according to UK guidelines.[15] This may include anti-seizure medication, rehabilitation of neurological deficits, medical treatment of other neurological symptoms, psychological support and MRI monitoring, determined by clinicians involved in each patient's care.[13]

### Ancillary and post-trial care
There are no provisions for ancillary care or care for participants after the trial ends. Because interventions in the CARE pilot trial are provided in standard clinical practice, aftercare will occur as standard practice.

### QuinteT recruitment intervention
#### Phase I
Before recruitment starts, the QRI researchers will qualitatively evaluate factors that may influence recruitment using focus groups comprised of healthcare professionals and PAG members. The QRI researcher will observe all CARE pilot trial management group (TMG) and trial steering committee (online supplemental material 2) meetings during protocol development.

During recruitment, the QRI researcher will use screening logs, recruitment consultation recordings, interviews with CARE researchers and participants, and observation of trial meetings to investigate recruitment obstacles.

#### Phase II
In parallel, findings from phase I will be presented to the CI and TMG and used to implement measures to improve recruitment and information provision.

### Outcomes
#### Primary outcome
We will estimate these measures of feasibility:
1. What proportion of the collaborating sites take part and recruit participants to the CARE pilot trial?

2. Can the investigators implement trial procedures correctly?
3. What proportion of screened patients are eligible?
4. What proportions of eligible patients are approached and randomised (and why are eligible patients not approached or not randomised)?
5. What is the distribution of participants between neurosurgery and stereotactic radiosurgery?
6. Do participants adhere to the allocated intervention and follow-up?
7. How complete are baseline, imaging and outcome data?
8. What are the outcome event rates?
9. How do the baseline characteristics, outcome event rates and differences between treatment groups compare to observational data about outcomes during medical management or after medical and surgical management?
10. What estimates of effect size/variability should be used in the design of the CARE definitive main phase trial?
11. What is the sample size required for a definitive trial to address the overall question over a 10-year follow-up?
12. Can the CARE pilot trial data describe care pathways, linked to health states and outcomes, to develop a robust economic model to evaluate cost-effectiveness in a CARE definitive main phase trial?

13. Which international research partners in other countries could contribute to the CARE definitive main phase trial?

### Primary clinical outcome

Intracranial haemorrhage or new persistent/progressive focal neurological deficit due to brain cavernoma or surgical management (neurosurgery or SRS), whether fatal (leading to death within 30 days of the outcome event) or non-fatal.

### Secondary clinical outcomes

1. Death not due to a primary clinical outcome.
2. Liverpool Seizure Severity Scale plus epileptic seizure frequency (number of seizures in the preceding 4 weeks, and attainment of 1 year seizure freedom).
3. Modified Rankin Scale (mRS) score.
4. National Institute of Health Stroke Scale Score (NIHSS; adult or paediatric).
5. 5-level EuroQol-5D questionnaire (EQ-5D-5L) in adults and EQ-5D Youth (EQ-5D-Y) in children.
6. Karnofsky Performance Status (KPS) scale in adults and Lansky Play-Performance Scale (LPS) in children.

We will also collect data to estimate health service use and healthcare and socioeconomic costs during the entire duration of follow-up.

**Table 1** Table of assessments

| Assessment | Identification and screening | Baseline visit | Within 3 months of baseline | 6-month local in-person follow-up | 6-monthly central follow-up |
|---|---|---|---|---|---|
| Assessment of eligibility | X | | | | |
| Screening end enrolment logs | X | | | | |
| Consent to recruitment conversation recordings | X* | | | | |
| Consent to qualitative interview | X | | | | |
| Recording of patient recruitment conversations | X† | X† | | | |
| Consent to randomisation | X‡ | X‡ | | | |
| Demographic, clinical, socioeconomic, medication and radiographic data | | X | | | |
| DNA sample | | X | | | |
| Provision of diagnostic brain imaging | | X | | | |
| Questionnaires | | X | | X | X |
| Randomisation | | X | | | |
| Cavernoma surgical management | | | X | | |
| Repeat brain MRI | | | | X | |
| Outcomes and adverse events | | | | X | X |
| Qualitative interview | | | X§ | | |

*Research teams will be asked to capture verbal consent to audiorecordings of recruitment conversations when the approach is made to the participant. If this is not possible at this time, consent may be captured during subsequent recruitment conversations.
†Recordings of recruitment conversations with patients should be captured (as requested) wherever the CARE pilot trial is discussed (illustrated here but not restricted to screening and baseline visit).
‡Consent to participation in CARE may be collected at the baseline visit or in advance, during the screening stage.
§Interviews with patients will take place within 3 months of being invited to take part in the trial.
CARE, Cavernomas: A Randomised Effectiveness pilot trial.

## Participant timeline

A detailed timeline for data collection is provided in table 1.

### Identification and screening

The research team will identify eligible patients from the UK and Ireland from multiple sources including data from hospital admissions, outpatient appointments, referrals, multidisciplinary team discussions, and routine brain imaging. Diagnoses may be made at any time during or before recruitment.

### Assessment of eligibility

Eligibility will be confirmed following discussion with the patient and a specialist in the type of treatment that is thought to be most effective for surgical management. Eligibility may be informed by multidisciplinary discussion.

### Baseline visit and consent

There is no specific time window for approaching eligible patients for consent. The baseline visit and consent meeting may be conducted remotely or in person, at the time of randomisation or shortly beforehand. The research team will collect a venous blood sample of up to 10 mL into an EDTA blood tube for DNA extraction during face-to-face visits.

### Surgical treatment

It is expected, but not mandated, that surgical management will be delivered within 3 months of randomisation. Adherence will be assessed remotely by the trial coordinating centre (TCC) at 3–6 months.

### Qualitative interviews

In-depth interviews will be conducted by the QRI researcher in a sample of eligible patients from a variety of sites who have been approached to participate in the trial, with priority given to those declining participation to explore reasons why. Purposive sampling will be used to identify patients. Interviews will take place within 3 months of the participation decision.

### Six-month follow-up visit

Participants will be asked to attend for their first 6-month follow-up visit in person to perform a brain MRI. Outcome questionnaires will be completed. If not collected at the baseline visit, a blood sample will be obtained.

### Six-monthly central follow-up

The TCC will subsequently perform 6-monthly postal follow-up, including completion of outcome questionnaires, after checking the patient's vital status with their general practitioner. A researcher will contact non-responders electronically.

### Long-term follow-up

We will ask study participants to consent to long-term follow-up, beyond the planned follow-up in the CARE pilot trial, including the use of routinely collected data in case the CARE pilot proceeds into a definitive main phase trial.

### Sample size

Approximately 240 people will be newly diagnosed with symptomatic brain cavernoma during 18 months of recruitment.[2] We aim for all of these patients to be screened, but if 10% are missed and 10% decline to participate, we expect research teams to identify 190 patients. In the ARUBA trial, 226/726 (31%) of the eligible patients approached were randomised, so we expect at least 60 patients with symptomatic brain cavernoma to be randomised in the CARE pilot trial.[16]

### Recruitment and consent

Eligible patients will be approached for recruitment during or following discussion with relevant secondary care specialists by research staff who are members of or affiliated to the clinical team and have undergone standardised training on trial-related procedures. An invitation letter may be sent to the patient in advance. Participant information leaflets and informed consent forms will be provided (online supplemental material 3). For children, participant information leaflets are available for children 0–5 years old, 6–10 years old and 11–15 years old. The patient or the parent/guardian will be given as much time as they require to consider the study information and ask questions. Written informed consent may be recorded in paper forms, electronic copies thereof or an online electronic consent form. Children aged 6–15 who can understand it will be given the option of providing assent.

When a child recruited into the trial reaches the age of 16 years (or 18 years old in Ireland) and is therefore competent to provide consent, they should be reconsented at their next 6-month follow-up review. No further data will be collected until a signed consent form has been received.

### Consent to be contacted for an interview exploring reasons for declining participation

Patients or their parents/carers who decline participation in the CARE pilot trial will be invited to consent to participate in an interview with a QRI researcher, exploring their experiences of being approached and invited to participate. Where parents/carers consent to take part in an interview, the child/young person may attend and contribute.

### Allocation

The consensus preference agreed between each patient and their clinician for neurosurgery or SRS, should randomisation allocate them to medical and surgical management, will be recorded at the baseline visit. If there is no clear preference and both are available, the patient will be randomly allocated to the type of surgical treatment they will receive, if allocated to surgical treatment (figure 1). Participants in these two strata will be assigned 1:1 to medical management or medical and

surgical management using permuted blocks. Allocation will be concealed until participants are enrolled and assigned using central web-based randomisation. Patients will be informed of their treatment allocation following randomisation.

### Blinding

Treatment allocation in the CARE pilot trial is not blinded, and is therefore open to participants, treating clinicians and research staff.

We will aim to keep outcome event assessors blind to treatment allocation. We will measure how often assessors are unblinded to treatment allocation during the process of event adjudication.

### Data collection

Demographic socioeconomic data and medical history will be collected at baseline visit alongside the following patient-reported questionnaires: EQ5D-5L (adults), EQ5D-3Y (children) and the Liverpool Seizure Severity Scale. Research staff will assess mRS score, NIHSS (adult or paediatric, if examined in person), KPS (adults) and LPS (children). Research teams will upload pseudoanonymised Digital Imaging and Communications in Medicine (DICOM) images of diagnostic brain imaging for validation by a senior neuroradiologist to confirm or refuse eligibility.

In-depth interviews will be conducted by a qualitative researcher within 3 months of their participation decision.

Participants will be asked to attend their 6-month follow-up visit in person for brain MRI to assess cavernoma presence and size, as a measure of treatment efficacy. As a minimum standard, T1-weighted, T2-weighted and haemsensitive gradient recalled echo or susceptibility-weighted imaging will be required. We will collect any other sequences performed. Images will be uploaded to the trial database and the radiology department at the participant's site will issue a clinical report. The local research team will record clinical outcome events since randomisation and the details of neurosurgery or SRS. Imaging studies performed because of an outcome event will be uploaded. The same patient reported questionnaires and standardised assessments used at baseline will be assessed at the first 6-month visit.

After this, the TCC will undertake 6-monthly postal, telephone or email follow-up. Questionnaires will ask about disability, health-related quality of life, the occurrence of primary or secondary clinical outcomes, SAEs, the occurrence of surgical management of the brain cavernoma (described above) and relevant concomitant medications (anti-seizure medication, propranolol, antiplatelet agents, anticoagulant agents and statins).

### Retention

We aim for >95% retention of participants at 6 months with <10% treatment group switches or loss to follow-up.

### Data management

Personal data will be processed by site research teams, the TCC at the University of Edinburgh (UoE) and qualitative research staff at the University of Bristol (UoB). Personal data will be stored securely at sites and the secure trial database, hosted on a UoE server. Brain imaging will be managed by the Systematic Management, Archiving & Reviewing of Trial Images Service at the UoE. Audiorecordings will be securely transferred by qualitative research team members onto a secure drive at the UoB for long-term storage and analysis. Audiorecordings will be labelled with the participant identification number but not identifiable patient details. Audiorecordings will undergo targeted transcription and editing to protect respondents' anonymity. This data will be managed using NVivo software and stored on encrypted UoB drives.

### Data analysis
#### Statistical analyses

In this pilot phase, analyses are descriptive only, and there will be no formal statistical tests. A detailed statistical analysis plan is described in online supplemental material 4. We will quantify the number and proportions (with 95% CIs to reflect their precision) of patients who are screened, eligible, approached, provide consent and are randomised.[17] We will construct a Consolidated Standards of Reporting Trials (CONSORT) diagram to summarise the distribution and progress of participants in the trial including the numbers of withdrawals.[18] We will report descriptively the following: the number and the proportion of the collaborating sites that take part and recruit participants to the CARE pilot trial; research teams' implementation of trial procedures measured by number and type of protocol deviation; the numbers of participants allocated to neurosurgery and SRS; adherence to the allocated intervention; completeness of follow-up that would be due at each 6-month interval; completeness of baseline, imaging and outcome data; the frequency of outcome events overall and in an intention-to-treat analysis keeping patients in the treatment group to which they were allocated during all available follow-up.

We will also compare descriptively the characteristics of eligible patients who are screened and do not participate in the CARE pilot trial to eligible patients who are randomised using the characteristics recorded on the screening logs to assess generalisability (external validity) and any recruitment bias. We will assess measures of functional outcome, to assess which has suitable statistical properties for use in a main phase trial (such as lack of floor/ceiling effects). We will assess whether such a measure (like the method we have used before[8]) would be more suitable as a primary outcome in place of intracranial haemorrhage.

#### QRI data analysis

The QuinteT researcher will analyse data using the SEAR framework to observe differences between sites in recruitment patterns as new sites open.[17 18] Descriptive analyses

will identify where patients are lost to recruitment and the reasons why.

Audiorecordings of recruitment conversations will be sought from a purposive sample of recruiting sites. The audiorecordings will explore information provision, management of patient treatment preferences and randomisation decisions to identify recruitment difficulties and improve information provision. Analysis will employ content, thematic and novel analytical approaches, including targeted conversation analysis and quanti-qual appointment timing.[19–22] Interview data will be analysed thematically using constant comparative approaches derived from Grounded Theory methodology.[23]

Findings from the QRI will be fed back to the CI and TMG, to determine a plan of actions to optimise recruitment.

### Health economics analysis

The full health economic analysis plan is in online supplemental material 5.[24 25] We will collect self-reported health service use and social/economic outcomes using bespoke question sets that will inform future economic analyses.[8 26] If data collection is confirmed as feasible, then a previously developed decision model will be updated and further developed to incorporate data collected within this study to provide a putative estimate of cost-effectiveness and its drivers.[27] In the context of the CARE pilot trial, the health economics objectives are to: (1) design and test an optimal mechanism for the capture of resource use and cost data in community National Health Service (NHS) settings, NHS secondary care, participants' out-of-pocket expenses and carer costs, (2) estimate expected effect size and variance of relevant outcomes including health-related utility and quality-adjusted life-years and (3) identify and measure the potential cost implications of surgical management of cavernomas.

We will measure health-related utility, healthcare-related resource use and costs using participant questionnaires before randomisation and at each follow-up time point.[20 28] These costs will be ratified by the study team through scrutiny of the patient pathway in both arms of the trials using available medical records to populate case report forms (CRFs). We will assign unit costs using standard national costing sources where available, or through consultation with relevant service business managers. Costs will be summarised from the perspectives of the NHS and personal social services, and wider society (including participants and their carers).

### Data monitoring
### Data monitoring committee

An independent data monitoring committee (DMC) has been established to oversee the safety of participants in the trial (online supplemental material 6). No formal interim analyses are planned during the conduct of the pilot trial.

### Adverse events

Participants will be instructed to contact their site research team if any symptoms develop at any time after being randomised. Participants will be asked about the occurrence of SAEs whenever contact is made with them between randomisation and the final central 6 monthly follow-up. SAEs may be identified via information from support departments, for example, laboratories. Only events which are clinical outcomes for the trial or are related to medical and surgical management and occur between randomisation and the final 6-month follow-up review will be recorded as AEs or SAEs. Only AEs or SAEs that are clinical outcomes or SAEs related to medical and surgical management will be recorded in the electronic CRF. If there is any doubt as to whether a clinical observation is an SAE, the event will be recorded.

When an SAE occurs, site research staff will review all documentation related to the event, assess whether an AE is an outcome in the trial and record all relevant information. If the AE is detected by central means of follow-up, the TCC will initiate the collection of this information but enlist the help of site research staff. This information will be reported to the ACCORD (Academic and Clinical Central office for Research and Development) Edinburgh Research Governance & Quality Assurance (QA) Office immediately or within 24 hours. The investigator will follow up each event until resolution. All reports sent to ACCORD and any follow-up information will be retained in the investigator site file. The sponsor is responsible for reporting SAEs that are 'possibly related' to the treatment allocation and 'unexpected', to the REC within 15 days of becoming aware of the event. The TCC will provide SAE line listings from ACCORD for circulation prior to DMC meetings.

### Audit

Investigators and institutions involved in the study will permit trial related monitoring and audits on behalf of the sponsor, ACCORD, research ethics committee review and regulatory inspection(s). Risk assessment, if required, will determine if an audit by the ACCORD QA group is required. If required, audit details will be captured in an audit plan.

## ETHICS AND DISSEMINATION
### Ethical conduct

The study will be conducted in accordance with the principles of the International Conference on Harmonisation Tripartite Guideline for Good Clinical Practice. Before the study begins all required approvals will be obtained, including that of the Yorkshire and The Humber—Leeds East Research Ethics Committee (REC; 21/YH/0046).

### Protocol amendments

Any changes in research activity, except those necessary to remove a hazard to the participant in the case of an urgent safety measure, must be reviewed and approved by the CI. Amendments will be submitted to the sponsor for review and authorisation before being submitted to the appropriate REC and local Research and Development team for approval.

## Open access

## Data sharing

Following publication of the primary results, a deidentified individual participant data set will be prepared for sharing purposes. All data requests should be submitted to the CI for consideration. Deidentified data collected during the QRI will be made available by the QuinteT research group to CAUK. Other individuals wishing to access deidentified QRI data may apply to an independent committee.

## Publication and dissemination

We will submit manuscripts to peer-reviewed journals for open access publication. We will present our findings at meetings of relevant professional associations.

## Insurance and indemnity

The University of Edinburgh has insurance in place for negligent harm caused by poor protocol design by researchers employed by the University of Edinburgh. Sites participating in the study will be liable for clinical negligence and other negligent harm to individuals taking part in the study and covered by the duty of care owed to them by the sites concerned. Sites which are part of the UK's NHS will have the benefit of NHS Indemnity.

### Author affiliations

[1]Centre for Clinical Brain Sciences, The University of Edinburgh, Edinburgh, UK
[2]Department of Clinical Neurosciences, Royal Infirmary of Edinburgh, Edinburgh, UK
[3]Royal Hallamshire Hospital, Sheffield, UK
[4]Neurosurgery, University Hospital of Wales, Cardiff, UK
[5]Aberdeen Royal Infirmary, Aberdeen, UK
[6]Hull Royal Infirmary, Kingston upon Hull, UK
[7]University Hospital Southampton NHS Foundation Trust Wessex Neurological Centre, Southampton, UK
[8]Edinburgh Clinical Trials Unit, The University of Edinburgh Usher Institute of Population Health Sciences and Informatics, Edinburgh, UK
[9]National Centre for Stereotactic Radiosurgery, Royal Hallamshire Hospital, Sheffield, UK
[10]Walton Centre for Neurology and Neurosurgery, Liverpool, UK
[11]Royal Preston Hospital, Preston, UK
[12]Bristol Royal Hospital for Children, Bristol, UK
[13]James Cook University Hospital, Middlesbrough, UK
[14]Developmental Neurosciences Department, Great Ormond Street Hospital for Children, London, UK
[15]The National Hospital for Neurology & Neurosurgery, London, UK
[16]Institute of Genetics and Cancer, University of Edinburgh, Edinburgh, UK
[17]Queen's Hospital, Romford, UK
[18]Queen Elizabeth Hospital, Birmingham, UK
[19]Clinical Neurosciences, University of Cambridge, Cambridge, UK
[20]Addenbrooke's Hospital, Cambridge, UK
[21]Centre for Clinical Neurosciences, Salford Royal Hospital Manchester, Salford, UK
[22]Alder Hey Children's Hospital, Liverpool, UK
[23]Charing Cross Hospital, London, UK
[24]Department of Neurosurgery, Atkinson Morley Wing, St George's Hospital, London, UK
[25]Department of Neurosurgery, Southmead Hospital, Bristol, UK
[26]Population Health Science, Bristol Medical School, University of Bristol, Bristol, UK
[27]King's College Hospital, London, UK
[28]Institute of Psychiatry Psychology & Neuroscience, King's College London, London, UK
[29]Cavernoma Alliance, Watlington, UK
[30]Newcastle University Translational and Clinical Research Institute, Newcastle upon Tyne, UK
[31]The Royal London Hospital, London, UK
[32]Sheffield Children's Hospital NHS Foundation Trust, Sheffield, UK

**Acknowledgements** We thank all members of the Patient, carer and public Advisory Group (PAG) for contributing to the development, design and delivery of this study.

**Collaborators** Royal Infirmary Edinburgh - Ioannis Fouyas, Allan MacRaild, Jessica Teasdale, Michelle Coakley, James Loan, Rustam Al-Shahi Salman, Paul Brennan, Drahoslav Sokol, Anthony Wiggins, Chandru Kaliaperumal, Mairi MacDonald and Sarah Risbridger; St.George's Hospital - Marios Papadopoulos, Siobhan Kearney, Ravindran Visagan, Ellaine Bosetta and Hasan Asif; Great Ormond Street Hospital - Greg James, Aswin Chari, Vijeya Ganesan, Martin Tisdall, Christin Eltze, Zubair Tahir and Sanjay Bhate; National Hospital Neurology and Neurosurgery - Patrick Grover, Azra Banaras, Sifelani Tshuma, Neil Kitchen, William Muirhead, Ciaran Scott Hill, Rupal Shah, Thomas Doke, Rebecca Hall and Sonny Coskuner; Royal Hallamshire Hospital - Andrew Bacon, Kirsty Harkness, Emma Richards, Jo Howe, Christine Kamara, Jonathan Gardner, Madalina Roman, Mary Sikaonga, Matthias Radatz, Julian Cahill, Alex Rossdeutsch, Varduhi Cahill, Imron Hamina and Kishor Chaudhari; Addenbrooke's Hospital - Adel Helmy, Liliana Chapas, Silvia Tarantino, Karen Caldwell, Mathew Guilfoyle, Smriti Agarwal, Daniel Brown, Sarah Holland and Tamara Tajsic; Alder Hey Hospital - Conor Mallucci, Anil Israni, Rachael Dore, Taya Anderson, Dawn Hennigan, Shelley Mayor, Laura O'Malley and Samantha Glover; Aberdeen Royal Infirmary - Pragnesh Bhatt, Janice Irvine, Sohail Majeed, Sandra Williams, John Reid, Annika Walch, Farah Muir and Eng Tah Goh; Queen Elizabeth Hospital, Birmingham - Tom Hayton, Arlo Whitehouse, Andrew McDarby, Michelle Bates, Rebecca Hancox, Edward White and Claudia Kate Auyeung; Birmingham Children's Hospital - William B Lo and Julie Woodfield; Southmead Hospital - Mario Teo, Jack Wildman, Kerry Smith, Elizabeth Goff, Deanna Stephens, Borislava Borislavova, Ruth Worner, Sandeep Buddha and Philip Clatworthy; Bristol Royal Hospital for Children - Richard Edwards, Karen Coy, Lisa Tucker, Sandra Dymond, Andrew Mallick, Rebecca Hodnett and Francesca Spickett-Jones; University Hospital Wales - Janneke van Beijnum, Paul Leach, Tom Hughes, Milan Makwana, Khalid Hamandi, Dymona McAleer and Belinda Gunning; Hull Royal Infirmary - Mihai Danciut, Emma Clarkson and Anna Bjornson; Walton Centre - Emmanuel Chavredakis, Debbie Brown, Giannis Sokratous, John Williamson, Cathy Stoneley, Andrew Brodbelt, Jibril Osman Farah and Sarah Illingworth; Charing Cross Hospital, London - Ramesh Nair, Sophie Hunter, Niamh Bohnacker, Rosette Marimon, Lydia Parker, Oishik Raha and Puneet Sharma; King's College Hospital, London - Daniel Walsh, Oliver Wroe Wright and Sabina Patel; Salford Royal Hospital - Dan Holsgrove, Danielle McLaughlan, Tracey Marsden, Francesca Colombo, Kathryn Cawley, Hellen Raffalli and Saba Raza-Knight; Manchester Children's Hospital - Ian Kamaly-Asl, Felicia Jennings, Nicola Phillips, Imedla Mayor, James Stewart, Dipek Ram, Rebecca Keeping, Grace Vassallo and Katie Hennessy; James Cook University Hospital - Nitin Mukerji, Emanuel Cirstea, Susan Davies, Venetia Giannakaki, Ammar Kadhim, Oliver Kennion, Md Moidul Islam, Lucie Ferguson and Manjunath Prasad; Royal Victoria Infirmary, Newcastle - Nicholas Ross, Beth Atkinson, Cheryl Webster, Michelle Fawcett, Vicky Slater and Saffnan Mohamed; Royal Preston Hospital - Nihal Gurusinghe, Saba Raza Knight, Terri-Louise Cromie, Allan Brown, Sonia Raj, Ruth Pennington, Charlene Campbell, Shakeelah Patel and Francesca Colombo; Queen's Hospital, Romford - Raghu Vindlacheruvu, Anthony Ghosh, Teresa Fitzpatrick and Lauren Harris; Sheffield Children's Hospital - Shungu Ushewokunze, Sarah Ali, John Preston, Carole Chambers and Mohammed Patel; Southampton General Hospital - Diederik Bulters, Ronneil Digpal, Winnington Ruiz, Mirriam Taylor, Divina Anyog, Katarzyna Tluchowska, Jackson Nolasco, Daniel Brooks, Kleopatra Angelopoulou, Bethany Welch and Nicole Broomes; Royal Stoke Hospital - Howard Brydon, Ida Ponce, Louis Taylor, Lucy Bailey, Mia Marsden, Claire Hudson, Angelene Cope, Jack Lee, Deepthy Blesson and Rachel Sutton; Leeds General Infirmary – Ian Anderson, Mary Kambafwile, Linetty Makawa, Jade McAndrew and Atul Tyagi; Royal London Hospital - Christopher Uff and Geetha Boyapati.

**Contributors** Conceptualisation: RA-SS and NK, supported by JJML, VG, PSH, KH, PJH, EK, SL, CM, ARN, MR, JS, AS, CT, JWa, DWh, and PW. Methodology: JJML, VG, PSH, KH, PJH, EK, SL, CM, ARN, MR, JS, AS, CT, JWa, DWh, PW, NK and RA-SS. Project administration: JJML, ABj, JvB, PB, ABu, NB, DB, JC, EC, FC, MD, RD, RJE, LFo, LFe, IF, VG, PG, NG, KH, LSH, TH, AH, DH, PJH, AI, EK, SM, CM, NM, RN, MCP, MR, AR, SR-K, MT, CT, JWa, DWh, DWa, PW, JWi, OWW, CU, SU, RV, NK and RA-SS. Funding Acquisition: RA-SS, supported by LFo, EK, and NK. Writing— original draft: JJML and RA-SS. Writing—review and editing: All. Supervision: RA-SS and NK.

**Funding** The CARE pilot trial is funded by the National Institute for Health and Care Research (NIHR128694), which requires publication of the trial protocol but had no other role in manuscript preparation or the decision to publish.

**Competing interests** PW declares institutional unrestricted educational grant funding for a stroke reperfusion course from Stryker, Penumbra and Medtronic. MR declares that he is a Senior Clinician of the National Centre for Stereotactic Radiosurgery.

**Patient and public involvement** Patients and/or the public were involved in the design, conduct, reporting, and dissemination plans of this research. Refer to the Methods section for further details.

**Patient consent for publication** Not applicable.

**Provenance and peer review** Not commissioned; peer reviewed for ethical and funding approval prior to submission.

**ORCID iDs**
James J M Loan http://orcid.org/0000-0002-6451-9448
Pragnesh Bhatt http://orcid.org/0000-0002-2145-4760
Anna Bjornson http://orcid.org/0000-0001-5616-6817
Alistair Bullen http://orcid.org/0000-0002-1655-6404
Diederik Bulters http://orcid.org/0000-0001-9884-9050
Julian Cahill http://orcid.org/0000-0003-0296-4412
Emmanuel Chavredakis http://orcid.org/0000-0001-7571-4233
Francesca Colombo http://orcid.org/0000-0002-2018-7779
Richard J Edwards http://orcid.org/0000-0001-8415-2180
Lucie Ferguson http://orcid.org/0000-0002-9011-4313
Vijeya Ganesan http://orcid.org/0000-0003-1864-6216
Patrick Grover http://orcid.org/0000-0002-7822-1239
Nihal Gurusinghe http://orcid.org/0000-0001-9706-0672
Peter S Hall http://orcid.org/0000-0001-6015-7841
Adel Helmy http://orcid.org/0000-0002-0531-0556
Peter J Hutchinson http://orcid.org/0000-0002-2796-1835
Steff Lewis http://orcid.org/0000-0003-1210-2314
Conor Mallucci http://orcid.org/0000-0002-5509-0547
Aileen R Neilson http://orcid.org/0000-0003-3758-0566
Marios C Papadopoulos http://orcid.org/0000-0001-9174-4176
Andrew Stoddart http://orcid.org/0000-0002-1958-3897
Mario Teo http://orcid.org/0000-0002-0051-3303
Carole Turner http://orcid.org/0000-0002-8297-4890
Julia Wade http://orcid.org/0000-0001-6486-6477
Daniel Walsh http://orcid.org/0000-0003-1274-3285
Christopher Uff http://orcid.org/0000-0001-9787-8001
Rustam Al-Shahi Salman http://orcid.org/0000-0002-2108-9222

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
