## [Reviewer comments · BMJ Open]

ARTICLE DETAILS

TITLE (PROVISIONAL)	Cavernomas A Randomised Effectiveness (CARE) pilot trial, to address the effectiveness of active treatment (with neurosurgery or stereotactic radiosurgery) versus conservative management in people with symptomatic brain cavernoma: Study protocol
AUTHORS	Loan, James; Bacon, Andrew; van Beijnum, Janneke; Bhatt, Pragnesh; Bjornson, Anna; Broomes, Nicole; Bullen, Alistair; Bulters, Diederik; Cahill, Julian; Chavredakis, Emmanuel; Colombo, Francesca; Danciu, Mihai; Dignpal, Ronneil; Edwards, Richard; Ferguson, Lucie; Forsyth, Laura; Fouyas, Ioannis; Ganesan, Vijeya; Grover, Patrick; Gurusinghe, Nihal; Hall, Peter; Harkness, Kirsty; Harris, Lauren S; Hayton, Tom; Helmy, Adel; Holsgrove, Daniel; Hutchinson, Peter; Israni, Anil; Kinsella, Elaine; Lewis, Steff; Majeed, Sohail; Mallucci, Conor; Mukerji, Nitin; Nair, Ramesh; Neilson, Aileen; Papadopoulos, Marios; Radatz, Matthias; Rossdeutsch, Alex; Raza-Knight, Saba; Stephen, J; Stoddart, Andrew; Teo, Mario; Turner, Carole; Wade, Julia; Walsh, Daniel; White, David; White, Phil; Wildman, Jack; Wroe Wright, Oliver; Uff, Christopher; Ushewokunze, Shungu; Vindlacheruvu, Raghu; Kitchen, Neil; Al-Shahi Salman, Rustam

This article was not externally reviewed at BMJ Open. Protocols that have been independently assessed prior to submission to BMJ Open are usually fast-tracked to publication on the grounds that further substantial changes will not be possible. This independent assessment will usually be external, independent review for both a substantial grant award from a non-commercial or government funder and ethics approval.